# Improved Functionality of Integration-Deficient Lentiviral Vectors (IDLVs) by the Inclusion of IS_2_ Protein Docks

**DOI:** 10.3390/pharmaceutics13081217

**Published:** 2021-08-06

**Authors:** Marina Cortijo-Gutiérrez, Sabina Sánchez-Hernández, María Tristán-Manzano, Noelia Maldonado-Pérez, Lourdes Lopez-Onieva, Pedro J. Real, Concha Herrera, Juan Antonio Marchal, Francisco Martin, Karim Benabdellah

**Affiliations:** 1GENYO, Centre for Genomics and Oncological Research, Genomic Medicine Department, Pfizer-University of Granada-Andalusian Regional Government, Health Sciences Technology Park, Av. de la Illustration 114, 18016 Granada, Spain; marina.cortijo@genyo.es (M.C.-G.); sabinash@biomed.au.dk (S.S.-H.); maria.tristan@genyo.es (M.T.-M.); noelia.maldonado@genyo.es (N.M.-P.); francisco.martin@genyo.es (F.M.); 2GENYO, Centre for Genomics and Oncological Research, Molecular Oncology Department, Pfizer-University of Granada-Andalusian Regional Government, Health Sciences Technology Park, Av. de la Illustration 114, 18016 Granada, Spain; lourdes.lopez@genyo.es (L.L.-O.); pedro.real@genyo.es (P.J.R.); 3Department of Biochemistry and Molecular Biology I, Faculty of Science, University of Granada, Avenida Fuentenueva s/n, 18071 Granada, Spain; 4Personalized Oncology Group, Bio-Health Research Institute (ibs Granada), 18016 Granada, Spain; 5Maimonides Institute of Biomedical Research in Cordoba (IMIBIC), 14004 Cordoba, Spain; inmaculada.herrera.sspa@juntadeandalucia.es; 6Department of Haematology, Reina Sofía University Hospital, 14004 Cordoba, Spain; 7Biomedical Research Institute (ibs. Granada), 18012 Granada, Spain; jmarchal@ugr.es; 8Biopathology and Regenerative Medicine Institute (IBIMER), Centre for Biomedical Research (CIBM), University of Granada, 18016 Granada, Spain; 9Department of Human Anatomy and Embryology, Faculty of Medicine, University of Granada, 18016 Granada, Spain; 10Excellence Research Unit: Modeling Nature (MNat), University of Granada, 18016 Granada, Spain

**Keywords:** IDLV, gene delivery, gene expression, gene editing, off-targets

## Abstract

Integration-deficient lentiviral vectors (IDLVs) have recently generated increasing interest, not only as a tool for transient gene delivery, but also as a technique for detecting off-target cleavage in gene-editing methodologies which rely on customized endonucleases (ENs). Despite their broad potential applications, the efficacy of IDLVs has historically been limited by low transgene expression and by the reduced sensitivity to detect low-frequency *off*-*target events*. We have previously reported that the incorporation of the chimeric sequence element IS2 into the long terminal repeat (LTR) of IDLVs increases gene expression levels, while also reducing the episome yield inside transduced cells. Our study demonstrates that the effectiveness of IDLVs relies on the balance between two parameters which can be modulated by the inclusion of IS2 sequences. In the present study, we explore new IDLV configurations harboring several elements based on IS2 modifications engineered to mediate more efficient transgene expression without affecting the targeted cell load. Of all the insulators and configurations analysed, the insertion of the IS2 into the 3′LTR produced the best results. After demonstrating a DAPI-low nuclear gene repositioning of IS2-containing episomes, we determined whether, in addition to a positive effect on transcription, the IS2 could improve the capture of IDLVs on double strand breaks (DSBs). Thus, DSBs were randomly generated, using the etoposide or locus-specific CRISPR-Cas9. Our results show that the IS2 element improved the efficacy of IDLV DSB detection. Altogether, our data indicate that the insertion of IS2 into the LTR of IDLVs improved, not only their transgene expression levels, but also their ability to be inserted into existing DSBs. This could have significant implications for the development of an unbiased detection tool for off-target cleavage sites from different specific nucleases.

## 1. Introduction

Over the last two decades, the field of gene therapy (GT) has been truly revolutionized in terms of safety and efficacy. GT has been used to successfully treat several otherwise incurable life-threatening diseases [1,2] which are generally targeted through the random introduction of genetic material into an individual’s genome [1]. However, the use of precise genome editing has expanded the field of GT, thanks to novel highly precise endonucleases (ENs) which enable the accurately targeted removal of damaged genes and deleterious sequences, as well as the insertion of therapeutic genes into specific genomic loci [3,4].

The main challenge for any GT-based approach is the development of an effective, precise, and non-toxic vector for delivery of genetic material and/or specific nucleases. Over the last two decades, both viral [5] and non-viral vectors [6] have been developed. Although new non-viral-based delivery methods have produced impressive results in preclinical studies, including low cytotoxicity and immunogenicity, the vast majority (almost 70%) of GT clinical trials carried out until now have used integrating and non-integrating viral-based techniques.

Although the vectors of choice of non-integrating systems are based on adeno associated vectors AAVs [7], concerns over their limited packaging capacity, transduction efficiency and their tendency to induce immune responses against AAVs have been raised [8]. In this respect, integration-deficient lentiviral vectors (IDLVs) have actually replaced rAAVs in protocols involving large DNA sequences [9]. IDLVs belong to the retroviral family of vectors whose members contain two single-strand RNA genetic material and an integrase (IN) mutation, which prevents the integration of their genetic material into the host cells [10,11]. These non-integrating viral vectors have been successfully used in basic, preclinical research, and clinical applications mainly due to their broad tropism and larger cargo capacity as compared to rAAVs [12]. IDLVs have been successfully used in several animal models in treatments including therapies for regenerative retinal disease [13] and hemophilia B [14,15], as well as to deliver endonucleases and other molecules in gene editing-approaches [16,17,18,19,20,21]. In addition, one of the most sought-after applications of IDLVs in the gene-editing field is related to their capacity to be captured by double-strand breaks (DSBs). The capacity of IDLVs to integrate into DNA DSBs has been used by several research teams for unbiased detection of off-target sites generated by endonucleases [22], such as zinc finger nucleases (ZFNs) [23], the CRISPR/Cas9 system and transcription activator-like effector nucleases (TALENs) [24]. However, the sensitivity of genome-wide DSB detection has been reported to be relatively low when combined with the CRISPR/Cas9 and TALEN systems which target Wiskott–Aldrich syndrome (WAS) and tyrosine aminotransferase (TAT) genes [24]. All this previous research work carried out by our group and others has stressed the need for further improvements in the properties of IDLVs, particularly in gene expression terms. However, no study has focused on improving IDLVs for DSB detection The improvement in gene expression levels was achieved through the inclusion of chromatin boundary elements (CBEs) and genomic elements based on scaffold/matrix attachment regions (S/MARs) [25,26,27,28]. This is principally due to the capacity of SAR elements to bind transcription factors, such as special AT-rich sequence binding protein 1 (STAB1) [29], nuclear matrix protein 4 (Nmp4), and the CCCTC-binding factor (CTCF) [30]. Our team and others have also investigated other elements based on the 1.2 kb 5′chicken β-globin insulator cHS4, resulting in increased episomal efficiency [31,32,33,34,35,36,37,38]. This is partly due to the interaction of cHS4 with the nuclear matrix though CTCF nuclear proteins [31]. A combination of both approaches enables chimeric genetic elements to be generated. For instance, our group recently designed the chimeric insulator IS2, which combines a 388 pb fragment composed of our MAR/SAR recognition signatures (MRS) pSAR2 with the genetic element cHS4_650 [38]. The insertion of this chimeric element into the 3′LTR of lentiviral vectors increases the transcriptional activity of IDLV episomes up to six-fold, but also causes a sharp reduction in the level of intracellular vector genomes, probably due to interference with the reverse transcription process which reduces particle formation [36,39]. The final impact of the IS2 element in IDLVs depends on the balance between these two effects on the target cells [36].

In this study, we explore different IDLV configurations harboring various IS2-based chromatin boundary elements (CBEs) in different locations with the aim of maximizing the positive effect on transcription, while preventing a decrease in intracellular IDLV episomes. However, none of the novel constructs managed to surpass the previous optimization of the IDLV containing the IS2 element within the LTR (SE-IS2 (*in*-LTR)), thus illustrating the need for two copies of both pSAR2 and HS4_650 elements flanking the viral vector in order to achieve the highest expression levels per episome. Finally, taking into account our previous data indicating that the IS2 element could reposition the IDLV episomes into DAPI-low nuclear compartments [36], we also investigated whether this repositioning could lead to improved efficacy of SE-IS2 (*in*-LTR) IDLVs captured by DSBs. Our results demonstrate that optimized IDLVs carrying IS2 sequences within the long terminal repeats (LTRs) may be a valuable alternative for not only IDLV-mediated gene expression but also for more sensitive detection of DSBs occurring naturally or mediated by off-target activity of ENs used in precise genomic medicine.

## 2. Materials and Methods

### 2.1. Cell Culture

Immortalized adherent HEK293T cells (ATCC^®^ CRL-11268) were maintained and cultured in Dulbecco’s Modified Eagle’s Medium (DMEM; Biowest) supplemented with 10% fetal bovine serum (FBS) at 37 °C and 10% CO_2_ atmosphere. Jurkat (ATCC^®^ TIB-152) cells were grown in suspension in RPMI-1640 (Gibco/Invitrogen) media supplemented with 10% FBS (Invitrogen) and 1% penicillin–streptomycin solution, maintained at 37 °C with 5% CO_2_. The human induced pluripotent stem cell (hiPSC) line PBMC1-iPS4F1 was generated, characterized, and cultured as described by Montes et al. [40]. Primary T cells and CD34+ hematopoietic stem cells were isolated from fresh or frozen apheresis products obtained from healthy donors at the Haematology Department of the Reina Sofía University Hospital (Córdoba, Spain) under written informed consent. Blood was 1:4 diluted in phosphate-buffered saline (PBS), and peripheral blood mononuclear cells (PBMCs) were collected after Ficoll separation (Lymphosep, Biowest). Human haemopoietic stem cells (hHSCs) were positively selected using CD34+ beads (Miltenyi Biotec) in an AutoMACS separator (Miltenyi Biotec) and were cultivated in StemSpan (Stemcell Technologies) supplemented with stem cell factor (SCF) (100 ng/mL), Flt3 Ligand (100 ng/mL), IL-6 (100 ng/mL), thrombopoietin TPO (100 ng/mL) from Peprotech, and UM171 (35 nM) from Stem Cell Technologies, as well as StemRegenin1 (0.7 mM, Caiman Chemicals) and PGE2 (10 mM) at 37 °C and 5% CO_2_. Pan T cells were isolated from the CD34 negative fraction using immunomagnetic beads and the MACSexpress Separator and were then cultivated in TexMACS medium supplemented with 20 UI/mL of IL-2 (Miltenyi Biotec).

### 2.2. Lentiviral Vector Constructs

The IS2 element, which combines the cHS4-650 fragment and pSAR2, was designed and synthesized as described elsewhere [38], while the plasmids SE [41] and SE-IS2 (*in*-LTR) [36] were characterized in a previous study. SE-SAR (*in*-LTR) was generated by inserting the pSAR2 element into the 3′LTR Bbs1 site of the SE plasmid. The SE-IS2 and SE-SAR plasmids were generated by inserting the IS2 and pSAR2 elements into the SE XhoI and SE KpnI sites, respectively. SE HS4_650 was generated through the insertion of the cHS4_650 element into the KpnI site in the SE plasmid.

### 2.3. Viral Production

Lentiviral vectors were generated by co-transfection of HEK-293T cells with the plasmid of interest together with the HIV packaging plasmid pCMVDRD8.74 (kindly provided by Howe S.J. at the Great Ormond Street Institute of Child Health, University College London, London, UK) and the p-MD-G plasmid encoding the vesicular stomatitis virus (VSV-G) envelope gene (http://www.addgene.org/Didier_Trono, accessed on 12 April 2020). Transfection was performed using LipoD293T (Signa Gen, Rockville, MD, USA) according to the manufacturer’s instructions. Viral supernatants were harvested 72 h after transfection and filtered through a 0.45 μm filter (Nalgene, Rochester, NY, USA).

### 2.4. Estimation of Viral Particles Available on the Packaging Cell Supernatant

To estimate the quantity of viral particles available on the packaging cell supernatant, we determined the number of transduction units per mL (TUs/mL) using a qPCR Lentiviral Titer Kit (ABM, Richmond, Canada) converting the amount of viral copies per mL (GC/mL) to TU/mL as indicated by the manufacturer.

The study was carried out in an Applied Biosystems 7500 Real-Time PCR System. Each sample was analysed in triplicate and an average amplification curve was generated.

### 2.5. Etoposide Exposure

The 293T cells were washed with Dulbecco’s phosphate buffered saline (DPBS, 1X; Biowest), counted and incubated overnight with different IDLVs. The media were washed and the cells were exposed to 1, 4, and 8 mM of etoposide for 2 or 8 h. The cells were then washed and eGFP expression levels were studied by flow cytometry over time.

### 2.6. CRISPR/Cas9 Gene Targeting

After 48 h of transduction with IDLVs, the Jurkat cells were nucleofected with CRISPR/Cas9 *ribonucleoproteins* (RNPs) directed to the TRAC of the TCR alpha gene using the CRISPR/Cas9 system, the Amaxa™ 4D-Nucleofector™ protocol, and the SF cell line kit (Lonza, Basel, Switzerland), together with the EO-115 program, according to the nucleofection protocol for Jurkat cells. The Jurkat cells were plated in 48-well plates and gene-editing efficiency was determined by flow cytometry five days after nucleofection.

### 2.7. Flow Cytometry

At various times, ranging from 72 h to 7 days after transduction, the different cell types were harvested and washed twice with fluorescence-activated cell sorting (FACS) buffer (PBS containing 2 mM EDTA and 2% FBS). In the case of human induced pluripotent stem (hiPS) cells, they were first dissociated using TrypleExpress (Life Technologies, Carlsbad, CA, USA), and cell suspension was stained with the pluripotent antibody marker Tra1-60 (PE, BioScience) for 20 min. Data were obtained using FACS Canto II flow cytometer and were analysed using FACS Diva software (BD Biosciences, San Jose, CA, USA). Enhanced green fluorescent protein (eGFP) expression was detected in the fluorescein isothiocyanate (FITC) channel. In the case of etoposide, 7AAD and PE Annexin V were used to measure dead and damaged cells. For the TCR gene-editing experiment we used anti CD3 PC7 antibodies against the CD3/TCR complex.

### 2.8. Viral Vector Genome Estimation

The IDLV DNA vector genomes in transduced cells were determined as follows: the transduced cells were lysed after three days in culture and genomic DNA was extracted using the QiAamp DNA Mini Kit (Qiagen). 60 ng of genomic DNA was submitted to quantitative PCR using ΔU3 Fw and PBS Rev primer pairs (See Appendix A), which enabled us to discriminate viral vector genomes from total IDLV DNA. The orientation of DU3Fw and PBS Rev only the PCR amplification of the reverse transcribed viral genome. On the other hand, the primers were appropriately designed to amplify the same amplicon in the case of the incorporation or not of the genetic element into the TLRs (See SUP3). As an internal control, we used primers for the human albumin locus (hAlb). The DNA of transduced cells was extracted 72 h after transduction using a QIAamp DNA Mini Kit (QIAGEN, Hilden, Germany); Applied Biosystems 7500 Real-Time PCR System software was used to carry out the reactions. Each sample was analysed in triplicate. The PCR reaction consisted of 40 cycles at 95 °C for 3 s and at 62 °C for 30 s, followed by the melting curve.

### 2.9. Estimation of IDLV Efficiency

The estimated relative efficiency of the different constructs was calculated as the percentage of eGFP+ cells or *cell* fluorescence intensity (MFI) divided by the amount of viral DNA genomes in the transduced cells for each treatment.

### 2.10. Statistical Analysis

All data are represented as means ± SEM. We used GraphPad Prism software (La Jolla, CA, USA) to carry out the unpaired two-tailed *t*-test.

## 3. Results

### 3.1. The Location and Composition of Chromatin Boundary Elements (CBEs) Greatly Affect IDLV Efficiency

We previously reported that the insertion of the IS2 element into the 3′LTR not only increased viral episome efficiency, but also drastically reduced viral DNA genome levels in transduced cells [36]. In this study, we therefore engineered novel IDLVs by placing the IS2 element or other elements based on IS2 modifications between the eGFP and the 3′LTR (SE-IS2), as shown in Figure 1A. We then evaluated their impact on episomal transcription levels and on intracellular IDLV yields in transduced cells. After producing these different IDLVs, we transduced 293T cells using an MOI 0.4, estimation based in Applied Biological Materials (ABM) lentiviral qPCR Titer Kit. Three days later, we analysed the eGFP expression levels of the different constructs (Figure 1B) in terms of the percentage of eGFP (Figure 1C, left), mean fluorescence intensity (MFI; Figure 1C, right) and the amount of viral genomes in transduced cells (Figure 1C, bottom). Our data show that the incorporation of IS2 into the LTR (SE-IS2 (*in*-LTR)) and pSAR2 outside the LTR (SE-SAR) significantly increased the percentage of eGFP positive cells and transgene expression levels (Figure 1B,C). However, the incorporation of IS2 outside the LTR (SE-IS2) only increased transgene expression levels (Figure 1C, right), with no positive effect on the percentage of eGFP positive cells. Meanwhile, the inclusion of pSAR2 within the LTR had no effect on the percentage of eGFP positive cells or on MFI expression levels (Appendix A). The insertion of IS2 into the LTR or downstream of eGFP reduced vector genomes in target cells by 62% and 29%, respectively, while the insertion of pSAR2 and cHS4-650 downstream of eGFP had no effect (Figure 1C, bottom). Altogether, the transcription efficiency of SE-IS2 (*in*-LTR) IDLV episomes was found to be 3–5 times higher than that of SE episomes (Figure 1D), which presumably counteracts the negative effect on vector genome levels in target cells. However, the improvement in SE-SAR is associated with a direct effect on transcription efficacy, without negatively affecting reverse transcription. However, the positive effect of pSAR on episomal transcriptional efficiency is lower than that of IS2 (Figure 1C, bottom), and the overall impact is slightly lower also (Figure 1D). Furthermore, the potential effect of the inclusion of IS2 in IDLVs backbone was analysed in three additional target cells with potential interest with regard for gene therapy and immunotherapy, such as primary T cells and CD34+ hematopoietic stem cells, or for basic research, in the case of Jurkat cells lines. The three-cell types were transduced with equal amount of SE and SE-IS2 (*in*-LTR) (MOI 0.4). Three days later, we analysed transgene expression level (MFI and percentage of eGFP+). The incorporation of IS2 resulted in significant increase in both parameters in the case of HSCs, when the data were relativized to the viral genomes inside the transduced cells. However, only a relative increase in the percentage of eGFP positive cells was observed in the case of Jurkat cells (See Appendix A).

### 3.2. SE-IS2 (in-LTR) Enhances SE-IDLV Efficiency in Human Induced Pluripotent Stem Cells (hiPSCs)

We next analysed the behaviour of different backbones engineered in human induced pluripotent stem cells (hiPSCs) due to their potential applications in basic science and advanced therapies. These hiPSCs were transduced, with an equal multiplicity of infection (MOI = 3) for each IDLV, and eGFP expression was analysed in TRA1-60^+^ hiPSCs. Transduction efficiency was found to be higher for IDLVs with IS2 in the LTR as compared to unmodified IDLVs (SE) (1.4 +/− 0.04-fold *p* = 0.05). Additionally, the inclusion of IS2 in the LTR enhanced transient eGFP expression levels in hiPSCs 72 h after transduction (1.17 +/− 0.02-fold *p* < 0.05) (Figure 2B,C). However, there was no improvement in IDLV features when the different elements were introduced outside the LTR. As expected, the inclusion of IS2 and other elements in the SE viral backbone resulted in different outcomes in terms of viral genomes in the target cells, whose effect was more pronounced for SE-IS2 (*in*-LTR) (70%). The reduction in viral genomes in target cells appears to depend on the size of the insert and the insertion site. The inclusion of cHS4_650 elements (650 bp) generated a reduction of over 50%, while the inclusion of SAR elements (388 bp) did not have any significant effect. In summary, we concluded that, as observed with 293T, the transcription efficiency of SE-IS2 (*in*-LTR) IDLV episomes in hiPSCs is up to 3–4 times higher than that of SE IDLV episomes (Figure 2D), which counteracts the negative effect on viral genome reverse transcription (Figure 2C, bottom).

### 3.3. Inclusion of IS2 Increases the Number of IDLVs Captured by DNA Double-Strand Breaks (DSBs)

IDLVs have previously been reported to have the ability to label DSBs induced by artificial endonucleases [23,42,43], ionizing radiation [44], or chemical agents [45]. IDLV-DSB labelling facilitates unbiased identification of potential off-targets attributed to artificial endonucleases, especially in those cells which are difficult to transfect. However, it should be taken into account that the IDLV off-target detection is less efficient than other methods, such as GUIDE-seq and Digenome-seq [46,47]. The inclusion of the IS2 element may recruit several proteins, including multifunctional proteins such as the CCCTC-binding factor (CTCF), which facilitate the integration of IDLVs into the DSBs, thus probably increasing sensitivity to DSB strapping [48,49,50]. We experimentally assessed whether the inclusion of IS2 enhances the capacity of SE-IDLVs to integrate into DSBs, thereby increasing the potential of IDLVs to act as off-target sensors. We transduced 293T cells using two IDLVs (SE and SE-IS2 (*in*-LTR)) with the same multiplicity of infection (MOI = 4) and then exposed the treated cells to 8 μM of etoposide for 8 h in order to avoid excessive cell death (See Appendix A). Next, we measured short-term, virtually transient eGFP expression (5 days) and long-term, integrative eGFP expression (36 days) (Figure 3B). As can be seen in Figure 3D, a sharp increase of 26.9% in eGFP-positive cells in response to etoposide was observed at the end of the experiment in cells transduced with SE-IS2 (*in*-LTR) IDLVs as compared to an increase of 6.70% with control cells transduced with SE IDLVs. These data are even more significant given the reduction in the starting viral genome in the cells in the case of the SE-IS2 (*in*-LTR) IDLVs, with a 70% lower level of viral genome than SE at the starting point (Figure 3C). We then sought to orthogonally validate the role of IS2 in IDLV trapping around the DSBs in a real gene-editing scenario. To this end, we transduced Jurkat cells with SE and SE-IS2 (*in*-LTR) IDLVs at a multiplicity of infection (MOI = 4) in order to obtain homogenous eGFP positive cells prior to nucleofection using CRISPR/Cas9 to target the T cell receptor alpha chain (TRAC) coding region (See Appendix A). We then evaluated the integration profiles of SE and SE-IS2 (*in*-LTR) IDLVs over time. Transduction of Jurkat cells, together with the nucleofection process, resulted in virtually a total population of eGFP positive cells. Although, as previously observed, the initial level of viral vector genomes inside the cells in SE-IS2 (*in*-LTR) IDLVs was found to be 75% lower than that for SE (Figure 4C), an equal amount of eGFP+ cells was reported 42 days after transfection, thus highlighting the high relative capture rates for SE-IS2 (*in*-LTR) IDLVs as compared to SE-IDLVs (Figure 4E). Altogether, these data strengthen our initial hypothesis that the inclusion of IS2 in the IDLV backbone increases the capture of IS2-containing IDLVs in DSBs. However, further study is needed to prevent the reduction in viral genome levels in cells due to LTR disruption by the IS2 insertion. These data, coupled with our finding that IS2 increases the IDLV insertion rate in randomly created DSBs induced by etoposide, indicate that IS2 may play an active role in attracting proteins involved in chromatin remodelling and/or DNA repair.

## 4. Discussion

The success of any gene therapy is largely associated with the development of gene-transfer vectors [2]. Over the last two decades, an impressive array of viral vectors has been developed for next-generation clinical trials [51] and to minimize the drawbacks associated with previous vectors [3]. Up to now, the most commonly used viral vectors in most clinical trials have been adenoviruses (AdVs), γ-retroviral vectors (RVs), adeno-associated vectors (AAVs), and lentiviral vectors (LVs) [1,7], with the latter presenting several attractive features, especially for applications where transgene integration is required. However, for other applications requiring transient expression for cell division or for stable expression in cells such as non-dividing neurons [52] and retinal cells [53], IDLVs are the ideal option due to their packaging capacity of up to 9 kb and their high tropism, which facilitate the transduction of a wide range of cell types. The major limitation of IDLVs as a delivery vehicle as compared to the integrative partner is the low expression level achieved, mainly due to the silencing they undergo inside the target cell nucleus [36,54]. The viral DNA of IDLVs undergoes a process of chromatinization immediately after entering the nucleus, which endows them with heterochromatin properties [20,55]. In recent years, we and other research groups have been attempting to remedy these shortcomings by incorporating genetic elements into the lentiviral backbone [36]. Through the insertion of the chimeric element IS2, which combines a synthetic SAR element (pSAR2) with a 650 bpβ-globin HS4 gene fragment (cHS4_650), we managed to prevent silencing and to enhance expression levels. However, as predicted, although a net increase in expression levels was observed, a significant decrease in the viral genome vector in the targeted cells also occurred, probably due to interference with the reverse transcriptase [36,39]. Thus, in order to increase transgene expression without affecting the number of viral genomes in transduced cells, in the present study, we reduced the size of the genetic elements used in the above-mentioned study which have been moved out of the LTRs. The desired improvement focuses on three features: (i) transgene expression; (ii) the amount of vector genomes entering the target cell; and/or (iii) the efficiency of the viral particle itself.

To improve the first feature without affecting the number of viral genomes in transduced cells, we included only a 400 bp pSAR component, which we compared to the original component IS2. To enhance their impact, both elements, which duplicate after the reverse transcription process, were incorporated into the LTR. The inclusion of IS2 (pSAR2 + HS4) in the LTR significantly affected reverse transcription, as shown by the viral DNA levels in the transduced cells, thus reducing the level of vector genomes in the cells. This decrease in viral genomes in cells was accentuated when IS2 was inserted into the LTR as compared to SAR. As previously noted, this effect is closely related to the efficiency of reverse transcription [34,56]. Thus, in the current study, we demonstrate that this reduction in viral genomes in transduced cells is directly correlated with the size of the insert, which is more pronounced with IS2 (1.2 kb) than with pSAR2 (400 bp). However, only the presence of IS2, not pSAR2, in the LTR offsets the negative effect on vector genome generation in target cells, indicating the key contribution of cHS4 derived elements to the IS2 effect.

The observed effect of IS2 was cell-type dependent, since no improvement was observed in absolute terms in the percentage of eGFP positive cells and in the transgene expression levels (MFI) in human hematopoietic stem cells (hHSCs), Jurkat cells, or even T cells, although the transcription efficiency of SE-IS2 (*in*-LTR) IDLV episomes only increased in the case of hHSCs.

Accordingly, and in order to minimize the negative impact on the viral genomes in the target cells, we explored an alternative positioning for insulator elements in the LV backbone. We placed the complete IS2 element and the separate pSAR2 and cHS4 elements outside the LTR and compared the positive effect of increased gene expression and the negative effect of decreased vector genome levels. As expected, the positioning of these genetic elements outside the LTR did not drastically disrupt viral packaging, which occurs with LTR manipulation. Unfortunately, we did not observe any enhancement in gene expression levels, which reinforced our hypothesis that both pSAR2 and cHS4 in duplicate are required to enhance IDLV gene expression, whose mode of action remains unclear. In our previous study, we found that the inclusion of IS2 enhances expression levels and modifies the positioning of the IDLVs in the nucleus [36]. This enables the viral vector to be placed in an independent domain to recruit regulatory proteins and to raise expression levels. The different components of these insulators cooperate in the recruitment of regulatory proteins; for instance, SAR elements interact with special AT-rich binding protein 1 (SATB1), nuclear matrix protein 4 (Nmp4) and the CCTC-binding factor (CTCF), which also interacts with the cHS4 element [57,58].

As mentioned previously, IDLV capture can be used in a predictive and truly unbiased manner to measure the specificity of any genome-editing technique [23,24,59]. Although it is possible to bioinformatically determine the off-target potential of a given gene-editing approach, these methods are frequently biased and require advanced next-generation sequencing technologies, as well as a highly sensitive detection capability [60,61,62]. Despite their potential use as a genome-wide tool, IDLVs are relatively insensitive given their integration deficiency and the in vitro positive selection requirement [42]. To overcome this limitation, we reasoned that the insertion of IS2 into the IDLV backbone could facilitate IDLV capture by DSBs and thus enhance sensitivity to DSB detection. To test this hypothesis, we performed two independent and complementary experiments: DSBs were created using both the CRISPR/Cas9 gene-editing tool and the topoisomerase II inhibitor etoposide; following these two treatments, part of the randomly induced DSBs was efficiently targeted by the viral genome present in the cell. As hypothesised, the presence of IS2 flanking the IDLV molecules increases the sensitivity of IDLVs to the DSBs. However, there are alternative genome-wide unbiased DSB identification tools such as GUIDE-seq, which is a highly sensitive assay for the detection of Cas9 off-target cleavage in living cells [46]. Although GUIDE-seq, with its detection of off-target sites with a frequency of 0.1% or lower, is highly sensitive, this method requires an efficient cellular transfection of the short dsODN tag, which makes it more challenging for poorly transfectable primary cells and even unfeasible for in vivo use. The high transduction efficiency of IDLVs in a broad range of cell types, including primary and iPS cells, the ease of DSB readouts, generally mediated by reporter gene expression, as well as the possibility of in vivo DSB detection, still make IDLVs an interesting platform for DSB detection.

However, although the inclusion of the IS2 element in the IDLVs will heighten their sensitivity for DSB detection, it could also limit their use as a transient vector system under certain circumstances and with specific cell types [10,63,64,65]. On the other hand, further studies are needed in order to elucidate the mechanism underlying the increase in DSB captures mediated by the incorporation of the IS2 into the LTR. Also, in order to exploit the potential applications of IDLV SE-IS2 (*in*-LTR) for unbiased detection of off-target cleavage sites, additional optimizations are required. In particular, we have been investigating how to maintain episomal repositioning without affecting the final viral genomes in the cells.

## 5. Conclusions

In the present study, we have developed an optimized IDLV system with a wide range of applications. This new system can fulfill different requirements for different applications, that require high transient gene expression either in vivo and in vitro. On the other hand, the modification mentioned increases the sensitivity of IDLVs to DSB detection. The optimized IDLV system could be used as a potential unbiased method for genome-wide off-target detection in genome-editing protocols. However, further studies based on a simultaneous verification of nuclear distribution of protein, such us SATB1, Nmp4, or CTCF, together with the IDLVs genomes and DSB sites, are required to elucidate the molecular machinery behind these observations and to optimize the system for increased sensitivity.

## Figures and Tables

**Figure 1 pharmaceutics-13-01217-f001:**
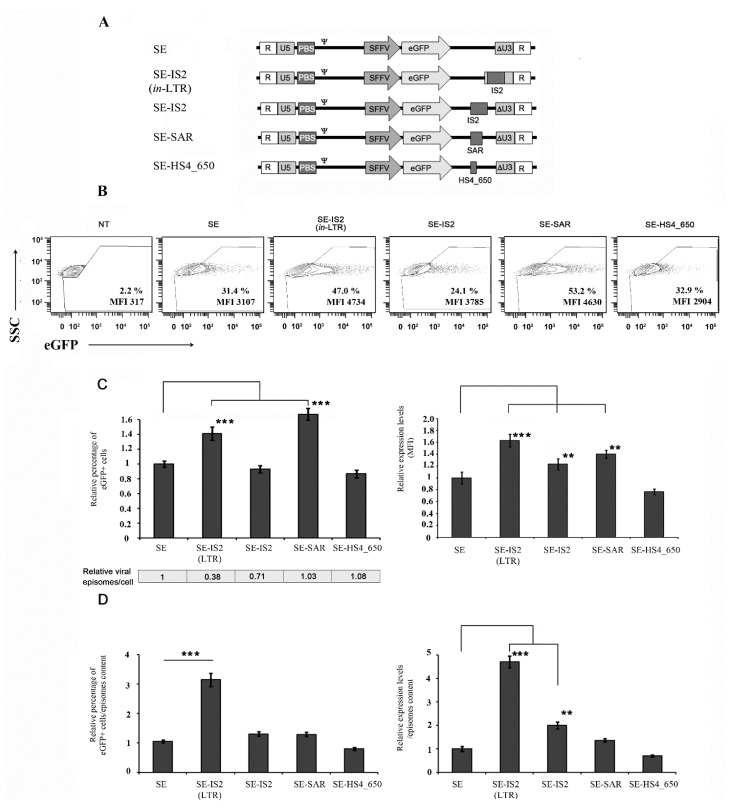
The IS2 element enhances expression levels inside the LTR, while the SAR element is more effective outside the LTR. (**A**) Schematic diagram of different constructs used in this study. (**B**) Representative plots of 293T cells transduced with SE, SE-IS2 (*in*-LTR), SE-IS2, SE-SAR, and SE-HS4_650 with an MOI = 0.4 and analysed 72 h after transduction, showing the percentage of eGFP+ cells and MFI. (**C**) Graphs showing the percentage of eGFP+ cells (left) and the eGFP expression levels (MFI, right) of 293T cells transduced with the different IDLVs relative to the values for SEs. Relative viral episomes in transduced cells are shown for each IDLV at the bottom of the graph on the left. (**D**) Graphs showing the estimated efficiency of the different constructs measured as the percentage of eGFP+ cells (left) or MFI (right) relative to the amount of viral episomes (as detailed in Materials and Methods). All graphs represent mean of at least three separate experiments, and error bars indicate standard error of the mean (SEM); *** = *p* < 0.001; ** = *p* < 0.05; and two-tailed unpaired Student’s *t*-test.

**Figure 2 pharmaceutics-13-01217-f002:**
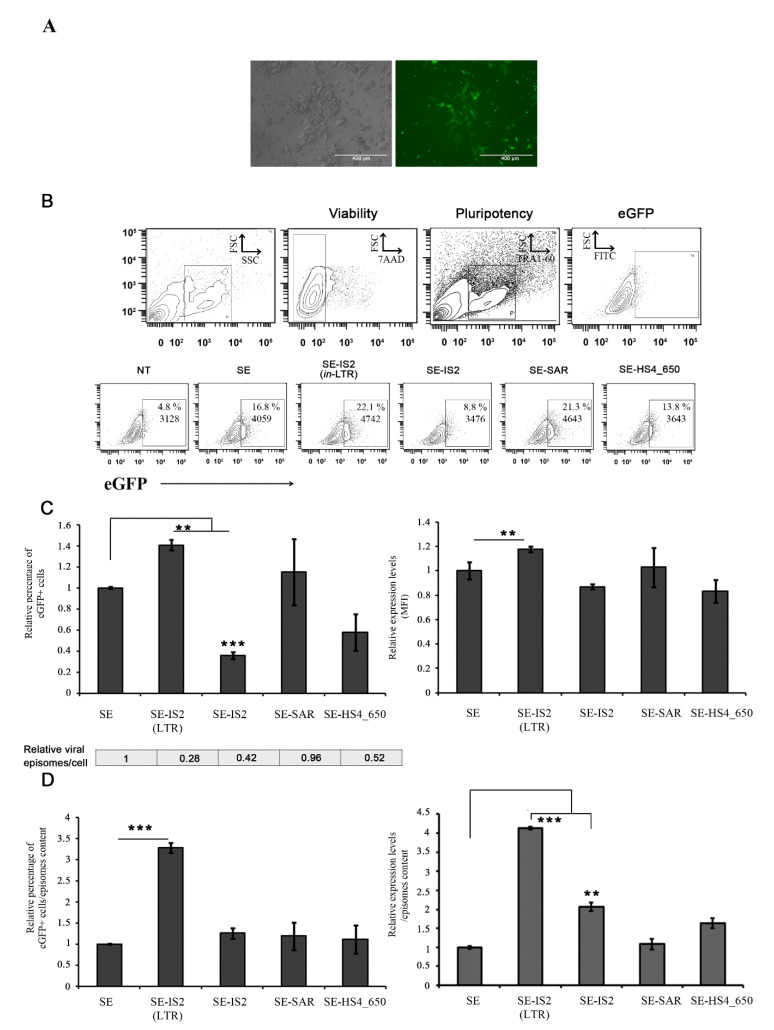
Effect of engineered genetic elements and position on IDLV-driven expression in hiPSCs. (**A**) Representative bright-field and EGFP fluorescence images of hiPSCs used in this study. (**B**) Representative plots of hiPSCs transduced with SE, SE-IS2 (*in*-LTR), SE-IS2, SE-SAR, and SE-HS4_650 with an MOI = 3 and analysed 72 h after transduction. (**C**) Graphs showing the percentage of eGFP+ cells (left) and the relative eGFP expression levels (MFI, right) of hiPSCs cells transduced with the different IDLVs relative to the values for SE. The relative viral episomes/transduced cell is shown for each IDLV below the graph on the left. (**D**) The graphs show the estimated efficiency of the different constructs relative to the amount of viral episomes (as detailed in Materials and Methods). All graphs represent mean of at least three separate experiments; error bars indicate standard error of the mean (SEM); *** = *p* < 0.001; ** = *p* < 0.05; and two-tailed unpaired Student’s *t*-test.

**Figure 3 pharmaceutics-13-01217-f003:**
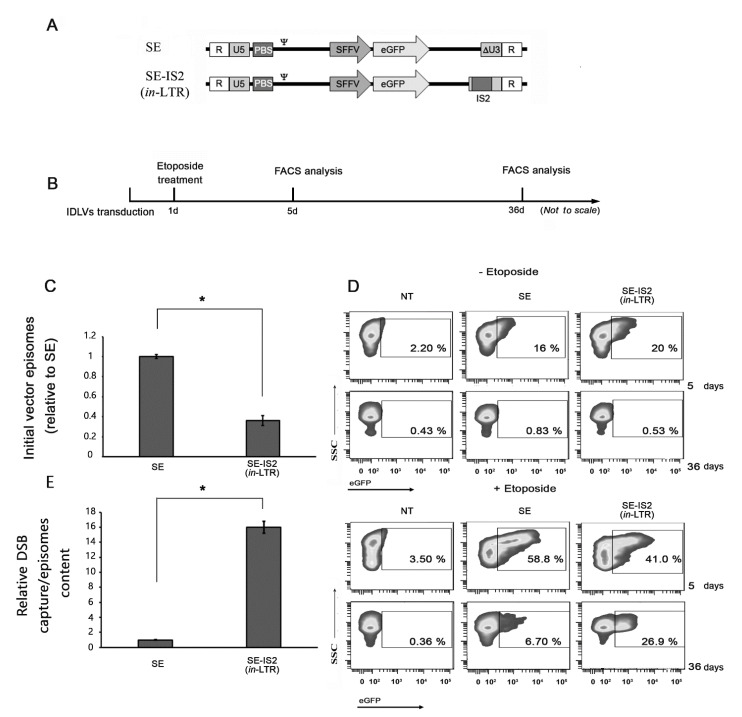
Presence of IS2 enhances the capture of IDLVs in DSBs. (**A**) Schematic diagram of the two constructs used in this study. (**B**) Workflow followed and experimental schema for IDLV gene trapping, 293T cell transduction with IDLV expressing GFP, followed by treatment with etoposide (8 µM). (**C**) Graph showing intracellular IDLV yield after transduction with the same multiplicity of infection (MOI = 4). (**D**) Representative experiment showing the increase in the percentage of eGFP expression, reflecting episomal relaxation due to etopside treatment, and the increased levels of eGFP labelling 36 d after transduction, which were further enhanced upon transduction with SE-IS2 (*in*-LTR). (**E**) IDLV capture by DSBs relative to the initial amount of viral genome episomes in the cells. The graphs represent mean of at least three separate experiments, while error bars indicate standard error of the mean (SEM); * = *p* < 0.01; two-tailed unpaired Student’s *t*-test.

**Figure 4 pharmaceutics-13-01217-f004:**
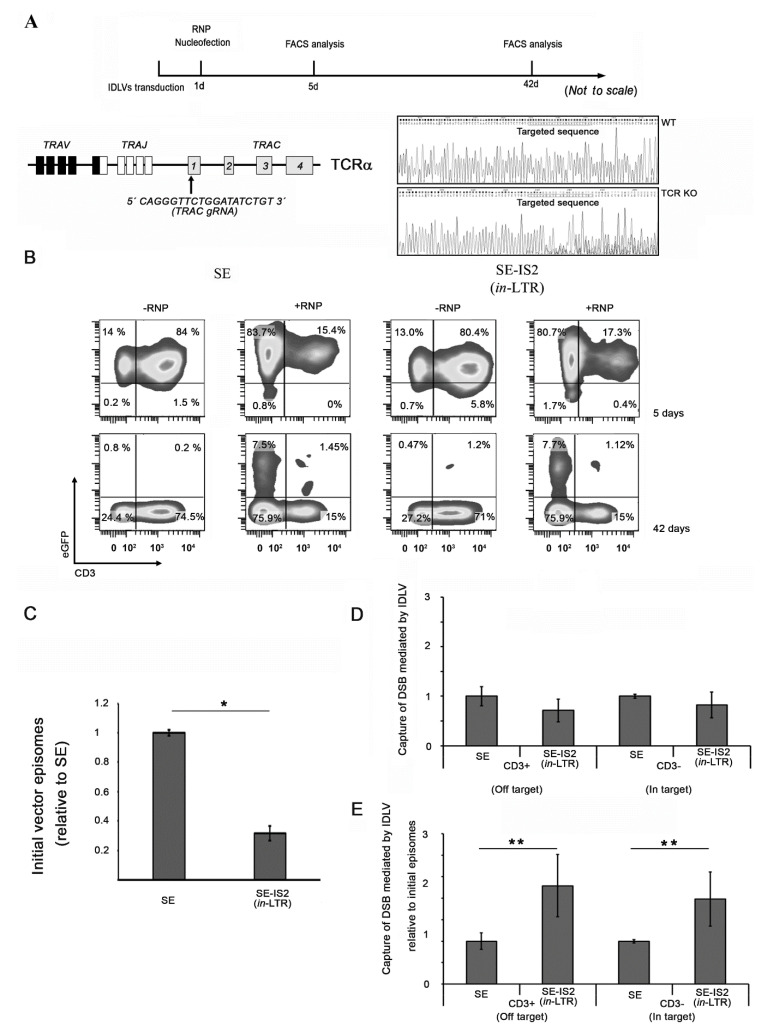
Presence of IS2 increases the sensitivity of IDLVs to DSB capture induced by sequence specific programmable nucleases. (**A**) (top) Workflow followed and experimental scheme for IDLV gene trapping; Jurkat cells were transduced with IDLV-expressing eGFP (MOI = 4), followed by electroporation with ribonucleoproteins (RNPs; gRNA + CAS9 proteins); (bottom left) CRISPR/Cas9-targeting TRAC locus showing the gRNA sequences used in this study; (bottom right) Sanger sequencing of bulk population showing the genome editing efficiency of the targeted locus. (**B**) Representative plots of Jurkat cells transduced with SE and SE-IS2 (LTR), nucleofected or not with RNPs targeting the T-cell receptor (TCR) locus 5 and 42 days after nuncleofection. (**C**) Relative vector DNA genomes in IDLV-transduced cells 72 h after transduction. (**D**) The graph shows the estimated capture efficiency of SE-IS2 (*in*-LTR) IDLVs relative to SE IDLVs either in the TCR locus (in target CD3^+^) or outside the TCR locus (off-target and random integration CD3^−^). (**E**) Capture efficiency after normalization to the intracellular IDLVs. All graphs represent mean of at least three separate experiments; error bars indicate standard error of the mean (SEM); * = *p* < 0.01, ** = *p* < 0.05; and two-tailed unpaired Student’s *t*-test.

## Data Availability

The data presented in this study are available on request from the corresponding author.

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
