# Peer review of "Improved Functionality of Integration-Deficient Lentiviral Vectors (IDLVs) by the Inclusion of IS2 Protein Docks"

_pharmaceutics, 2021, doi:10.3390/pharmaceutics13081217_

Round 1

Reviewer 1 Report

The manuscript "Improved functionality of integration-deficient lentiviral vectors (IDLVs) by the inclusion of IS2 protein docks" is really interesting from the technical point of view and although I have no concerns about the scientific point of view, it requires profound editing. Along with the manuscript, we can found plenty of odd long spaces between words, the figures are all together, which must be placed before or after each result section. Moreover, reference numbers are duplicated in the reference section.

Author Response

Reviewer 1

Review of “Improved functionality of integration-deficient lentiviral vectors (IDLVs) by the inclusion of IS2 protein docks”

The manuscript "Improved functionality of integration-deficient lentiviral vectors (IDLVs) by the inclusion of IS2 protein docks" is really interesting from the technical point of view and although I have no concerns about the scientific point of view, it requires profound editing. Along with the manuscript, we can find plenty of odd long spaces between words, the figures are all together, which must be placed before or after each result section. Moreover, reference numbers are duplicated in the reference section.

Authors Answer:

We thank the referee for the kind words and advices. We have submitted the paper to a deep editing process, in order to eliminate any  odd long spaces accordantly. Additionally, we  have gone through reference list in order to eliminate any inaccuracies or typographical errors.

Reviewer 2 Report

Improved functionality of integration-deficient lentiviral vectors (IDLVs) by the inclusion of IS2 protein docks  is an interesting study where the authors have followed-up on their previously published article in Nucleic acids (Sabina Sánchez-Hernández et al., 2019). In the current work, they have explored new IDLV configurations bearing multiple elements based on IS2 modifications. They have shown that the insertion of IS2 at the 3’ LTR showed promising results on efficient transcription and also detection of double stranded breaks.  The manuscript is well-written and the study is an important addition to IDLV literature.

I have few concerns/suggestions outlined below:

  1. The introduction could be shortened, some concepts are again mentioned in the discussion. For eg., gene therapy and use of viral vectors.
  2. It is important to include the method of viral titration and also the MOI used for each experiment
  3. For viral production, the authors have mentioned that they have used HIV packaging plasmid and pCMVDRD8.74 and pMD-G for packaging the IDLVs. Does pCMVDRD8.74 not have packaging components (gag,pol) already (maybe ‘and’ should be removed).
  4. What antibody was used for TCR profiling after gene editing?
  5. Explain the Sup fig 2 data, rather than stating data was cell-type dependent
  6. Maybe the authors could discuss how to further improve/explore how IS2 increases IDLV insertion rate in DSBs?
  7. The authors have mentioned in the abstract ”After demonstrating a DAPI-low nuclear gene repositioning of IS2 containing episomes... “ . Where is this data?
  8. In their previous studies, the authors have shown that the Inclusion of the Insulator improves Expression Levels in an HDAC-Independent Manner. It would be interesting to see the results of IS2 modifications used in the current study with and without HDAC inhibition.  
  9. Fig 1: A space between B and C panels would be recommended as "eGFP" should be more aligned towards panel B.
  10. Fig 2: Panel B- label the top panel flow plots
  11. Overall, the font sizes of figures can be made bigger for clarity.

Author Response

Reviewer 2

Improved functionality of integration-deficient lentiviral vectors (IDLVs) by the inclusion of IS2 protein docks  is an interesting study where the authors have followed-up on their previously published article in Nucleic acids (Sabina Sánchez-Hernández et al., 2019). In the current work, they have explored new IDLV configurations bearing multiple elements based on IS2 modifications. They have shown that the insertion of IS2 at the 3’ LTR showed promising results on efficient transcription and also detection of double stranded breaks.  The manuscript is well-written and the study is an important addition to IDLV literature.

Authors Answer:

We thank the referee for the kind words

Critique:

Reviewer 2. Comment 1

The introduction could be shortened, some concepts are again mentioned in the discussion. For eg., gene therapy and use of viral vectors.

Authors Answer:

Thank you for the advice.  we have shortened the introduction according to  the reviewer comment (Line 62-67).

Reviewer 2. Comment 2

It is important to include the method of viral titration and also the MOI used for each experiment.

Authors Answer:

We would like to thanks the referee for such valuable observation, and we apologize for not be precise in the definition of the viral titration and MOI used in each experiment. We used the ABM's Lentiviral qPCR Titer Kit that measures the amount of transduction units (TU) per milliliter. ABM provide an equation to calculate the titer that contains a specific coefficient determined in-house (ABM´ property) to convert the value from Genome copies per ml (GC/ml) to Transduction units (TU/ml); as such, the final titer value obtained is in TU/ml. The new manuscript has been revised to clarify this point in results. We have modified the manuscript (line 172) to clarify the methodology employed.

We have also modified the text in order to clarify the MOI used in each experiment (line: 234, 258, 270, 299, 310, 332, 345, 357 and 369)

Reviewer 2. Comment 3

For viral production, the authors have mentioned that they have used HIV packaging plasmid and pCMVDRD8.74 and pMD-G for packaging the IDLVs. Does pCMVDRD8.74 not have packaging components (gag,pol) already (maybe ‘and’ should be removed).

Authors Answer:

We apologize for this mistake. As the reviewer point out, the pCMVDRD8.74 is the HIV packaging plasmid contain the gag, pol, tat, and rev genes. We have modified the text accordantly (line 162/pg5).

Reviewer 2. Comment 4

What antibody was used for TCR profiling after gene editing?

Authors Answer:

We apologize, if our explanation may not have been sufficiently clear at this point. The genome editing efficiency was evaluated and confirmed by flow cytometry with antibodies against the CD3/TCR complex. Specifically, we use the antibodies CD3 PC7, we have modified the text in the revised manuscript of further clarify this point (line 200).

Reviewer 2. Comment 5.

Explain the Sup fig 2 data, rather than stating data was cell-type dependent.

Authors Answer:

We thank the reviewer comment.  We have included new text in the revised manuscript to explain Sup fig 2 (lines 252-261).

Reviewer 2. Comment 6.

Maybe the authors could discuss how to further improve/explore how IS2 increases IDLV insertion rate in DSBs?

Authors Answer:

We thank the reviewer advice and have included further discussion to cover this aspect (line 484-485). To explore how the IS2 element increases IDLV insertion rates, we could perform immune FISH to detect potential co-localization of IDLVs +/- IS2 and different proteins involved in chromatin structure or DSB repair to which the IS2 contain several binding sites. Once detected the potential players, we can disrupt or increase the amount of binding sites for the protein that co-localized with the IS2-IDLVs.

Reviewer 2. Comment 7.

The authors have mentioned in the abstract ”After demonstrating a DAPI-low nuclear gene repositioning of IS2 containing episomes... “ . Where is this data?

Authors Answer:

We are sorry about this confusion, the DAPI-low nuclear gene repositioning of the IS2 containing episomes was hitherto described in our previously published article in Nucleic acids (Sabina Sánchez-Hernández et al., 2019).

Reviewer 2. Comment 7.

In their previous studies, the authors have shown that the Inclusion of the Insulator improves Expression Levels in an HDAC-Independent Manner. It would be interesting to see the results of IS2 modifications used in the current study with and without HDAC inhibition.  

Authors Answer:

We agree with the reviewer. It would be interesting to verify that the increase in Expression Levels rate of the SE-IS2 (out), SE-SAR and SE-HS4-650 are not affected by HDAC as observed previously with the SE-IS2 (LTR). However, since the new construct have the same IS2 element but in a different location (SE-IS2 (out)) or have part of it (both SAR and HS4-650 are contained in the IS2), we consider it is not fundamental for the main conclusions of this manuscript.

Reviewer 2. Comment 8.

Fig 1: A space between B and C panels would be recommended as "eGFP" should be more aligned towards panel B.

Authors Answer:

Thank you for the advice. We have made changes in figure 1 accordantly.

Reviewer 2. Comment 9.

Fig 2: Panel B- label the top panel flow plots

Authors Answer:

Thank you for the advice. We have made changes in figure 2 accordantly.

Reviewer 2. Comment 10.

Overall, the font sizes of figures can be made bigger for clarity.

Authors Answer:

Thank you for the advice. We have made the required changes, when it possible,  in all figures accordantly.
